# PIAS Factors from Rainbow Trout Control NF-κB- and STAT-Dependent Gene Expression

**DOI:** 10.3390/ijms222312815

**Published:** 2021-11-26

**Authors:** Fabio Sarais, Sophia Kummerow, Ruth Montero, Henrike Rebl, Bernd Köllner, Tom Goldammer, Bertrand Collet, Alexander Rebl

**Affiliations:** 1Research Institute for Farm Animal Biology (FBN), Institute of Genome Biology, 18196 Dummerstorf, Germany; sarais@fbn-dummerstorf.de (F.S.); kummerow@fbn-dummerstorf.de (S.K.); tom.goldammer@fbn-dummerstorf.de (T.G.); 2Friedrich-Loeffler-Institut (FLI), Federal Research Institute for Animal Health, Institute of Immunology, 17493 Greifswald, Germany; ruth.montero@fli.de (R.M.); bernd.koellner@fli.de (B.K.); 3Department of Cell Biology, Rostock University Medical Center, 18057 Rostock, Germany; henrike.rebl@med.uni-rostock.de; 4Faculty of Agriculture and Environmental Sciences, University of Rostock, 18057 Rostock, Germany; 5Laboratoire Virologie et Immunologie Moléculaires, Institut National de Recherche Pour L’agriculture, L’alimentation et L’environnement (INRAE), Université Paris-Saclay, 78350 Paris, France

**Keywords:** CRISPR/Cas9, innate immunity, immune regulation, JAK-STAT signalling, NF-κB

## Abstract

Four ‘protein inhibitors of activated STAT’ (PIAS) control STAT-dependent and NF-κB-dependent immune signalling in humans. The genome of rainbow trout (*Oncorhynchus mykiss*) contains eight *pias* genes, which encode at least 14 different *pias* transcripts that are differentially expressed in a tissue- and cell-specific manner. *Pias1a2* was the most strongly expressed variant among the analysed *pias* genes in most tissues, while *pias4a2* was commonly low or absent. Since the knock-out of Pias factors in salmonid CHSE cells using CRISPR/Cas9 technology failed, three structurally different Pias protein variants were selected for overexpression studies in CHSE-214 cells. All three factors quenched the basal activity of an NF-κB promoter in a dose-dependent fashion, while the activity of an Mx promoter remained unaffected. Nevertheless, all three overexpressed Pias variants from trout strongly reduced the transcript level of the antiviral Stat-dependent *mx* gene in ifnγ-expressing CHSE-214 cells. Unlike *mx*, the overexpressed Pias factors modulated the transcript levels of NF-κB-dependent immune genes (mainly *il6*, *il10*, *ifna3*, and *stat4*) in ifnγ-expressing CHSE-214 cells in different ways. This dissimilar modulation of expression may result from the physical cooperation of the Pias proteins from trout with differential sets of interacting factors bound to distinct nuclear structures, as reflected by the differential nuclear localisation of trout Pias factors. In conclusion, this study provides evidence for the multiplication of *pias* genes and their sub-functionalisation during salmonid evolution.

## 1. Introduction

The signalling through Janus kinases (JAK) and signal transducers and activators of transcription (STAT) [1,2] transfer a wide range of information from the membrane to the nucleus of eukaryotic cells [3,4]. Upon the stimulus-dependent activation of specific cytokine receptors, the four mammalian JAK proteins (JAK1, JAK2, JAK3, and TYK2) become activated and phosphorylate several other associated proteins, including themselves, other receptor chains, and STAT factors [5] (Appendix A). The seven mammalian STAT proteins (STAT1, STAT2, STAT3, STAT4, STAT5a, STAT5b, and STAT6) dimerise after phosphorylation and translocate into the nucleus, where they bind to cognate DNA elements. These binding sites are often in close proximity to the response elements of nuclear factor-κB (NF-κB).

The NF-κB/Rel family of transcription factors comprises five members (p65/RelA, RelB, c-Rel, and p50/NF-κB1, p52/NF-κB2) in most vertebrates [6]. Together, STATs and NF-κB/Rel factors co-regulate a variety of inflammatory genes [7], including cytokines and, in particular, interferons (IFN) [8]. During evolution, several regulatory mechanisms have evolved to fine-tune both the intensity and the duration of cytokine signalling [9,10,11,12], including the ‘protein inhibitors of activated STATs’ (PIAS; Figure 1) [13,14]. The human PIAS family is comprised of four members: PIAS1, PIAS2/PIASx, PIAS3, and PIAS4/PIASy. By contrast, only one PIAS protein is present in the lancelet (*Branchiostoma* sp.) [15].

In fish, three orthologues of human PIAS1, PIAS2, and PIAS4 have been identified [16]. *Pias1* and *Pias4* are present as a pair of paralogue genes (a and b) in most fish species, but (pseudo-/allo-)tetraploid families, such as Salmonidae or Cyprinidae, are expected to encode additional *Pias* ohnologues.

The vertebrate PIAS proteins constitute a subfamily of E3 SUMO (small ubiquitin-related modifier) ligases. SUMO ligases tag their substrates post-transcriptionally with small ubiquitin-related modifiers to control the activity of transcription factors, affect the localisation of certain proteins, and inhibit or activate enzymes [17]. The characteristic functional motifs and domains are largely conserved across the vertebrate PIAS members. The N-terminal SAP (scaffold attachment factor A/B/acinus/PIAS) domain recognises and binds to A/T-rich DNA regions [18]. The PINIT motif [19] and two adjacent nuclear localisation signals (NLSs) allow for the retention of PIAS in the nucleus. A RING-finger-like zinc-binding domain (Siz/PIAS RING finger, SP-RING) [20] is followed by a SUMO-interacting motif (SIM) [21], and both are required for SUMO-protein ligase activity and interaction with other proteins. At the C-terminus, PIAS proteins contain a serine- and threonine-rich (S/T) region of unknown function [22].

PIAS factors employ the SUMOylation mechanism to negatively regulate the transcription of target genes by (i) blocking the DNA-binding activity of transcription factors [23,24,25]; (ii) recruiting histone deacetylases and other regulators, which modulate chromatin compaction [26]; and (iii) isolating transcription factors in specific subnuclear structures that are enriched with corepressor complexes [27,28]. Most studies have focused on the interaction of mammalian PIAS and STAT factors, but PIAS proteins can also block the activity of the NF-κB/Rel factors [15]. PIAS1 prevents the binding of RelA/p65 to the RELA response element of a distinct panel of mainly pro-inflammatory NF-κB-dependent genes [23,29], whereas PIAS3 suppresses the interaction of RelA/p65 and its coactivator CREB-binding protein (CBP) [30,31]. In addition to their inhibitory activities, PIAS proteins can also positively regulate the activity of certain transcription factors [32] (Appendix A). Mammalian PIAS4, for instance, may activate NF-κB by SUMOylating the inhibitor of NF-κB kinase subunit gamma (IKBKG) [33].

The only study on the three PIAS4 orthologues of zebrafish (*Danio rerio*) revealed that Pias4a regulates the Ticam1 (TIR domain-containing adaptor molecule 1)/Trif-dependent Ifn and NF-κB pathways [34]. Comparatively more studies have revealed the involvement of Stat proteins in controlling teleostean Ifn pathways [35,36,37]. The present study comparatively studied the structure, expression, and function of the three PIAS orthologues in salmonid fish and detected indications for their sub-functionalisation.

## 2. Results

### 2.1. PIAS Genes Are Present as Multiple Gene Copies in Rainbow Trout and Chinook Salmon

We searched the NCBI gene database for orthologues encoding ‘protein inhibitors of activated STAT’ in two *Oncorhynchus* species: rainbow trout (*O. mykiss*) and Chinook salmon (*O. tshawytscha*). The PIAS-encoding genes *pias1*, *pias2*, and *pias4* were available, whereas a *pias3* orthologue was absent not only in the *Oncorhynchus* spp., but in all teleostean species. Notably, all three *PIAS* orthologues in the *Oncorhynchus* spp. were present as multiple gene copies on eight chromosomes in rainbow trout (Table 1), similar to Chinook salmon. A synteny analysis allowed us to deduce the ancestry of the pias genes (Figure 2). *Pias1* genes are characteristically located adjacent to *skor1* across the analysed vertebrate species, except for Chinook salmon (Figure 2a). The common adjacency to *morf4l1* and *uaca* indicates that the two *pias1* genes on chromosomes 26 and 30 of rainbow trout are ohnologues, *pias1a1* and *pias1a2*. *Pias1* on chromosome 2 is flanked by a different set of genes, suggesting that this is a paralogue of *pias1a1* and *pias1a2* and should be termed *pias1b*. The *pias2* genes on chromosomes 6 and 11 of rainbow trout are both in close proximity to *npr3*, *arid3a*, and *bmp3*, indicating a common origin (Figure 2b). The two ohnologous genes should be referred to as *pias2a1* and *pias2a2*. We note, in this context, that none of the genes flanking the mammalian *PIAS2* are located near their teleostean orthologues. All *pias4* genes are flanked by *map2k2* and *onecut* across the analysed teleost species (Figure 2c). The two *pias4* genes on chromosomes 4 and 5 of rainbow trout are both adjacent to *zbtb7a* and *eef2* and should be termed *pias4a1* and *pias4a2*.

*Pias4* on chromosome 8 is flanked by a different set of genes, including *lingo3*, and should thus be termed *pias4b*. The annotation of the *pias* genes from Chinook salmon was less obvious, as some genes have not yet been localised.

The lengths of the sequences coding for the three *pias* paralogues ranged from 1422 bp (*pias4b*) to 2130 bp (*pias1a1*, LOC110527003) in rainbow trout (Table 1).

Although the *pias-*encoding sequences in Chinook salmon are in part shorter, the orthologous sequences from rainbow trout and Chinook salmon share a high level of identity of up to 96% (Figure 3). The identity between salmonid *pias* genes and their human orthologues ranges between 35% and 76% (Figure 3a–c). *PIAS1* shares the highest degree of identity across vertebrate species (Figure 3a), while *PIAS4* is less well conserved across vertebrates (Figure 3c).

The overall architecture of PIAS proteins is well conserved across vertebrates (Figure 4).

All Pias variants identified in rainbow trout contain the PINIT motif, followed by the SP-RING and SIM domain. A KxKELYRRR motif (amino acid (aa) residues 56–64, Pias1a) and the nucleoplasmin nuclear targeting signal (KK(x)_9_KK) are signatures of nuclear proteins (aa 373–384, Pias1a; aa 358–369, Pias1b) and characterize the Pias orthologues from *Oncorhynchus* sp. (Appendix A). Some of the analysed variants diverge significantly from the canonical Pias structure. The N-terminal SAP domain is present in all Pias1 aa sequences (Figure 4a, Appendix A), but is absent in Pias2a1 (Figure 4b, Appendix A) and the two Pias4b ohnologues (Figure 4c, Appendix A) from rainbow trout. While a centrally located NLS is included in the SP-RING domain of all Pias proteins of rainbow trout, the N-terminal located NLS is absent in Pias2a1 and all Pias4 paralogs of rainbow trout. In addition, all Pias4 protein variants of rainbow trout lack the S/T-rich region (Figure 4c, Appendix A), which is present in Pias1 and Pias2.

Notably, the *Oncorhynchus* Pias1b sequences are elongated at the N-terminus by a stretch of six amino acid residues, but this is absent in the mammalian counterpart (Appendix A). Conversely, more than 30 amino acid residues at the N-terminus extend the human Pias2 sequence (Appendix A). No signal peptide has been predicted for any Pias sequence from *Oncorhynchus* sp., but several disorder regions are present and are mostly located at positions conserved across vertebrate species (Appendix A).

The three-dimensional reconstructions of the three Pias factors from rainbow trout (Figure 4d–f) illustrate the well-conserved tertiary structure of the particular domains. The characteristics of the domains that are critical for nuclear localisation, SUMOylation, and zinc finger activity are located in a similar configuration to those observed in their human counterparts [38].

### 2.2. The Expression of Pias Genes Is Tissue and Cell-Type Specific

We used qRT-PCR to quantify the transcript levels of ten *pias* gene and transcript variants in all the selected tissues (brain, gills, head kidney, liver, muscle, spleen, and trunk kidney) of adult rainbow trout (Figure 5a–g). The *pias* gene variants did not show a uniform expression pattern; instead, they were regulated in a tissue-specific manner. *Pias1a2* was the dominantly expressed *pias* transcript variant (0.9 to 2.9 × 10^3^ transcripts/µg RNA) in gills, liver, spleen, head kidney, and trunk kidney, while its paralogue *pias1b* was the dominant *pias* transcript in the brain (2.7 × 10^3^ transcripts/µg RNA) (Figure 5a). In muscle, the overall *pias* transcript levels were low, with *pias4b.2* as the major *pias* transcript (1.4 × 10^3^ transcripts/µg RNA) (Figure 5e).

As a complement to our qPCR measurements in the whole tissues, we also analysed the levels of the various *pias* transcript variants in four cell models: (i) the secondary cell line CHSE-214, a model for functional studies of our *pias* constructs; (ii) freshly isolated head-kidney cells from rainbow trout; (iii) a non-myeloid (mAb21N) fraction enriched with T- and B-lymphocytes, natural killer-like cells, and thrombocytes; and (iv) a myeloid (mAb21P) fraction enriched with dendritic cells, granulocytes, and monocytes/macrophages from the head kidney of rainbow trout (Figure 5h–k). Strikingly, the model cells revealed considerable transcript levels of *pias4a2* (>1 × 10^3^ transcripts/µg RNA), which was more or less absent in the whole tissues previously analysed. By contrast, *pias4a1*.2 was constitutively expressed in all the selected tissues but was almost absent in the cell models. The *pias4a1.1* transcript variant was absent or present at an undetectable level (<20 transcripts/µg RNA in the liver) across all analysed tissues and cell models. The expression of *pias4b* was also not detected in the CHSE-214 cells (Figure 5h). Despite this observation, the expression of the *pias1* and *pias2* transcript variants in CHSE-214 was roughly comparable to that of the head kidney, with *pias1a2* as the most strongly expressed variant (3.3 × 10^3^ transcripts/µg RNA). The primary head-kidney cells had over 50% higher expression of various *pias* genes (*pias1a1*, -*1b*, -*2a1*, -*2a2*) compared with CHSE-214 cells. The expression of *pias* genes was even stronger in the non-myeloid cell fraction (up to 1.2 × 10^5^ *pias* transcripts/µg RNA).

In the head-kidney cells and the non-myeloid cells, *pias2a2* was the most strongly expressed *pias* gene (>6.0 × 10^3^ transcripts/µg RNA), while the *pias4a2* gene was most strongly expressed in the myeloid cell fraction (5.6 × 10^4^ transcripts/µg RNA).

We also quantified the expression of three *stat1* genes (Figure 6a–k), which are targeted by activated Pias factors. Overall, the qPCR expression data provided no obvious evidence for co-expression of *stat* and *pias* genes, as the transcript levels of *stat1a1* and *stat1a2* uniformly dominated over the *stat1b1* levels across all the tissues and cell models investigated.

In general, the expression levels of the three *stat1* genes were highest in the leukocytes enriched with myeloid cells (1.1 × 10^5^ *stat1a1* transcripts/µg RNA), as observed for *pias* expression. Remarkably, the transcript level of the three *stat* genes exceeded the *pias* levels, in general, by approximately 10-fold. Moreover, the expression levels of the three *stat1* genes were roughly 10 times lower in CHSE-214 cells than in the primary head-kidney leukocytes.

### 2.3. Pias1 and Pias2 Are Located in the Nucleus of Model Cells and Interact with NF-κB to Alter Transcriptional Responses

The prominent expression of *pias1* and *pias2* in CHSE-214 cells (Figure 5h) suggested that a CRISPR/Cas9 strategy might knock out both genes in the CHSE-214-derived cell line CHSE-EC [39]. The specific short guide (sg) RNA targets were located 20 bp upstream of the neighbouring ‘protospacer adjacent motif’ in exon 2 on *pias1*, in the first exon on *pias2*, and in the *mEGFP* (monomeric enhanced green fluorescence protein) gene. Cell sorting selected 48 GFP-negative single cells (Figure 7a), but only about 30 single cells generated a clonal cell line for each experiment. We sequenced the first 300 nt of the mEGFP gene and the sgRNA target exon of both *pias* genes. All sequenced clones had a deletion of one nucleotide in the ORF of the *mEGFP* gene, resulting in a frameshift and gene disruption (Figure 7b). Despite this observation, none of the mEGFP-negative clonal cell lines were mutated for the *pias1* or *pias2* genes. This suggests that the CRISPR/Cas9 system basically worked in CHSE-EC cells, and we can only speculate that a knock-out of the genes of interest might have been lethal for the cell [8]. In-silico analysis at DepMap Portal (https://depmap.org/; accessed on 1 October 2021) supports this assumption, since human *PIAS2* and *PIAS4* genes were designated as strongly selective.

Since the knock-out of both *pias1* and *pias2* genes failed, we selected three *pias* sequences for overexpression in the CHSE-214 cell model based on their explicit structural differences (cf. Figure 4a,b). Pias1 (709 aa) largely corresponds to the human orthologue, and Pias2 (508 aa) lacks the SAP domain and one of the two NLS, while the truncated Pias1 variant (233 aa) contains only the SAP domain and the respective NLS. Confocal microscopy revealed that full-length Pias1 and Pias2 from rainbow trout (both flagged with green fluorescent protein, GFP) apparently shared the same subcellular localisation in the nucleus of unchallenged or stimulated cells (Figure 8a,b) as the human orthologue [23,40]. Remarkably, Pias1 was more homogeneously distributed across the nucleus, while Pias2 was located at distinct nuclear spots.

The functions of the three Pias variants were explored by transient overexpression in CHSE-214 cells in separate approaches, together with a luciferase-reporter construct under the control of either (i) a rainbow trout *mx* promoter with an interferon-stimulating response element [41,42] or (ii) a human NF-κB-responsive *ELAM* (endothelial cell-leukocyte adhesion molecule) promoter [43]. Co-transfection with a construct expressing the ifnγ (encoded by the *ifng* gene, LOC100136413; NCBI acc. #AY795563) from Atlantic salmon, *Salmo salar*, simulated antiviral immune responses involving the production and secretion of ifnγ and the subsequent activation of the jak-stat pathway. The endogenous ifnγ synthesis induced a significant four-fold increase in the NF-κB promoter activity and approximately a two-fold increase in the *mx* promoter activity (both set as 1.0 in Figure 8c,d) over cells not transfected with the ifnγ-expression construct.

In cells that co-expressed one of the three Pias factors and the *mx*-reporter construct, we observed that neither different concentrations of transfected Pias-expressing vectors nor the additional endogenous expression of ifnγ affected the reporter-gene activity (Figure 8c). By contrast, we detected a significantly reduced reporter-gene activity in cells co-expressing one of the three Pias factors and the *ELAM*-reporter construct (Figure 8d). The use of 50 ng and more of Pias1-expression vector, 500 ng and 2000 ng of Pias2-expression vector, and 2000 ng of truncated Pias1-expression vector reduced the reporter-gene effectivity down to at least 0.6 (with *p* < 0.01). In other words, 50 ng of vectors expressing either Pias2 or truncated Pias1 were ineffective. The cells producing endogenous ifnγ underwent a similar Pias-induced effect, but this effect was not significantly different from that observed in unstimulated cells that did not produce ifnγ.

Having established that the three analysed Pias factors affected NF-κB activity, we used qPCR to test whether the overexpression of Pias1, truncated Pias1 or Pias2 modulates the transcription of a panel of early immune genes, which are regulated by the crosstalk of NF-κB and stat factors (cf. Figure 1b).

Ifnγ expression resulted in a ~two-fold increase in *mx* transcript levels and a slight, but significant, reduction in the *ifna1* transcript levels (Figure 9a). The use of 500 and 2000 ng of Pias1 and Pias2 both caused a 2.1- to 4.2-fold decrease in *il6* transcripts. The highest concentration of Pias2 also caused a significant decrease in *il10*, *ifna3*, and *stat4* copies by 1.6- to 2.0-fold. Of note, *tgfb* was the only gene that was significantly (two-fold) upregulated in concentration after the addition of 2000 ng of Pias1 (Figure 9a). The addition of 50 ng or 500 ng of Pias1 or truncated Pias1 from rainbow trout only slightly modulated the concentration of *mx* transcripts, but the addition of 2000 ng of one of the three Pias factors caused a strong decrease of more than ten-fold. The use of 500 ng of Pias2 also significantly (7.4-fold) lowered the number of *mx* transcripts. Altogether, 2000 ng of Pias2 downregulated the transcript levels of eight studied genes (*cxcl8*, *tgfb*, *il10*, *il6*, *ifna1*, *ifna3*, *stat4*, and *mx*) to a greater or lesser extent; therefore, we determined the mRNA concentration of four additional genes (*gata3*, *mmp9*, *socs1*, and *tp53*; cf. Figure 1b, [7]). The use of 2000 ng of Pias2 also reduced the transcript levels of *mmp9*, *gata3*, and *socs1* by 1.9- to 3.3-fold, while the levels of *tp53* remained unchanged (Figure 9b).

## 3. Discussion

PIAS proteins interact with more than 60 different proteins linked with transcriptional processes [23,24,25] and fine-regulate not only the STAT-dependent pathway but also NF-κB-signalling. Four different Pias proteins are encoded in humans, whereas PIAS3 has been lost in fish. In the present study, we found eight Pias-encoding genes in *Oncorhynchus* sp., and these likely result from two additional genome duplications in teleosts [44] and salmonids [45]. These eight genes encode at least 14 different pias transcripts. We speculate that the fish-specific and salmonid-specific duplications expanded the pias gene family in fishes by additional members, thereby compensating for the loss of PIAS3. The question that remains is why the number of Pias proteins in trout is still twice as high as that in humans.

In previous studies, we inspected the diversity of ohnologue and paralogue genes in salmonid fish that encode immune inhibitors [14]. We repeatedly found that structural modifications of the protein may fundamentally impact its function [46,47]. However, all three Pias1 proteins from rainbow trout strongly resemble the prototypical PIAS architecture of their human orthologue. This structural conservation suggests that the three Pias1 proteins from rainbow trout are functionally homologous to their counterparts in other vertebrates. One of the two Pias2 proteins from trout also resembles human PIAS1, while its paralogue and one of the Pias4 paralogues lack the SAP domain and one NLS. Similarly, all Pias4 proteins from trout lack the ST-rich region (as their mammalian counterparts), which is vital for the pleiotropic interactions associated with SUMO. Confocal microscopy provided evidence that Pias1 (with its SAP domain and two NLS) and Pias2 (without a SAP domain and with only one NLS) differentially localise in the nucleus. The ‘dot-like signatures’ of Pias2 from trout have also previously been observed for Pias4 from zebrafish [34]. Mammalian PIAS proteins interact with different (sets of) other (transcription) factors [48,49]. The dissimilar localisation of Pias1 and Pias2 from trout probably results from their cooperation with different sets of proteins that bind to distinct nuclear structures. The LxxLL motif, in particular, has been established to interact with nuclear receptors and co-receptors [49]; this motif is present in Pias1 but missing in Pias2 from rainbow trout.

Expression profiling of the individual *pias* variants from trout provided further evidence of their sub-functionalisation. While the *stat1* transcripts were expressed in relatively similar ratios to each other across the tissues analysed, the *pias* transcripts showed a rather tissue-specific expression pattern. Although *pias1a1* is the predominantly expressed gene in most tissues and cell fractions, the *pias4* transcripts, which do not encode an ST-rich region, dominated in muscle and in cell fractions enriched with dendritic cells, granulocytes, and monocytes/macrophages. The non-myeloid cell fraction contained mostly *pias2a2* transcripts, which are structurally quite similar to the *pias1a1* transcripts. The non-myeloid fraction is enriched with lymphocytes and thrombocytes and may also contain hematopoietic progenitor cells. The murine PIAS1 pathway has been reported to regulate self-renewal and differentiation of hematopoietic stem cells [50]. Therefore, the role of *pias1* and *pias2* in immunity and haematopoiesis remains an open topic for research in trout. The myeloid cell fraction expressed the highest levels of *pias* transcripts in general, thereby underpinning the importance of the Pias factors in immunophysiology.

Functional in vitro studies on Pias proteins in a non-mammalian model have only been carried out for Pias4 from zebrafish [34] to date. Therefore, we restricted our studies to one representative of Pias1 and Pias2 from rainbow trout and a truncated variant of Pias1 for comparative purposes. Since our expression studies on the salmonid fish cell model revealed that all transcript variants of both *pias1* and *pias2* were abundantly detectable, our aim was to knock out both factors using CRISPR/Cas9 technology. Since this approach was not successful, we were left with overexpression, a well-established method [51], to gain first insights into the function of trout Pias factors.

Both Pias1 and Pias2 from rainbow trout reduced the basal activity of NF-κB in unstimulated cells to a similar extent and in a dose-dependent fashion. This was not expected, as St2/Il1rlL1, another established inhibitor of NF-κB signalling, failed to modulate the basal level of NF-κB in our previous in vitro experiments in rainbow trout [52]. However, a study on murine PIAS1 has pointed to its potential to reduce NF-κB activity in a dose-dependent manner [23]. Murine PIAS1 directly interferes with the binding of NF-κB p65 to its corresponding response elements [48], and this mechanism is apparently conserved in fish. Interestingly, both Pias1 and Pias2 were similarly efficient in lowering the level of activated NF-κB, even though Pias2 lacks the SAP domain with the intrinsic LxxLL motif. A point worth mentioning in this context is that the expression of a mammalian PIAS3 mutant with point mutations in the LxxLL motif also did not interfere with the NF-κB activity [31].

In contrast to NF-κB-dependent promoters, Pias1 and Pias2 from rainbow trout did not modulate the activity of a trout *mx* promoter. Mx is a potent effector of antiviral defence [53]. We observed that overexpressed *pias1*, *pias2* and truncated *pias1* from trout strongly reduced the transcript level of *mx* in ifnγ-expressing cells. Apart from this effect, the truncated Pias1 variant did not modulate the transcript levels of any of the selected immune genes. Pias1 and, to a lesser extent, Pias2 lowered the transcript level of *il6*. In this regard, we note that the JAK-STAT signalling pathway is also known as the IL6 signalling pathway [54], as IL6 activates the cascade and thus also regulates its own expression. In contrast to the full-length and truncated Pias1 variants, Pias2 also reduced the transcript levels of several other NF-κB-dependent genes (*ifna3*, *stat4*, and *il10*) and, beyond those, the STAT-dependent genes *socs1*, *gata3*, and *mmp9*. These apparently different efficiencies in expression regulation likely reflect another consequence of the structural differences between Pias1 and Pias2 from trout. Previous studies have demonstrated that the overexpression of PIAS proteins enhanced the SUMOylation of nuclear receptors [55]. Since Pias1 and Pias2 from trout contain the required SP-RING and SIM domains, both should be capable of transferring SUMO proteins and thus altering the activity of transcription factors. The SAP domain is crucial for the translocation of transcription factors to the nuclear periphery [18], and for this reason, only Pias1, but not Pias2, from trout should be capable of an alternative regulatory mechanism that does not involve SUMO tags.

In conclusion, this study provides evidence for the multiplication of *pias* genes and their sub-functionalisation during salmonid evolution. For the functional analysis, we largely relied on the widely used CHSE-214 cell line. We note, in this regard, that this model cell is characterised by certain immunocompetence [56,57], although it cannot represent the complex interactions of different cell populations that tailor immune responses in vivo. For this reason, the CHSE cell line is a helpful tool, but it alone is not sufficient to map out the multiple functions of the Pias factors. Subsequent studies may investigate the influence of Pias proteins from trout using different immune cell subsets and in vivo.

## 4. Materials and Methods

### 4.1. Sampling and Cell Sorting

Rainbow trout (*O. mykiss*) were obtained from a local commercial fish farm ‘Forellenzucht Uthoff GmbH’, Neubrandenburg (Germany). Four fish were euthanised using an overdose of benzocaine (100 mg/L, Sigma-Aldrich/Merck, Steinheim, Germany) in compliance with the relevant European guidelines on animal welfare (Directive 2010/63/EU on the protection of animals used for scientific purposes) and were approved by the institute’s ethics board (approval ID: FLI 28/17). For the preparation of leukocyte suspensions, head kidneys were homogenized separately in 5 mL of 1% newborn calf serum (NCS)/phosphate-buffered saline (PBS) buffer (FB buffer). Cell suspensions were centrifuged at 4 °C at 290 g for 5 min and then resuspended in 3 mL FB. A Percoll gradient to discard erythrocytes was prepared as described previously [58]. One million leucocytes were labelled for 30 min at 4 °C with the monoclonal antibody 21 (mAb21, [59]), which recognises cells from a myeloid lineage. Thereafter, the cells were washed by centrifugation at 300× *g* and 4 °C for 5 min in 700 µL MACS Buffer (Miltenyi Biotec, Bergisch Gladbach, Germany). The pellet was resuspended in 200 µL of secondary antibody solution containing anti-mouse IgG-conjugated magnetic beads (Miltenyi Biotec), followed by a 30 min incubation at 4 °C and a final washing step. The cells were resuspended in 500 µL MACS buffer (Miltenyi Biotec, Germany) and placed into ice-cold racks to perform the magnetic separation in the autoMACS Pro Separator (Miltenyi Biotec). The sorting was conducted using the Possel_S program. After the separation, the enriched (mAb21-positive) cell fraction consisted of >95% myeloid cells, and the depleted (mAb21-negative) fraction consisted mostly of B- and T-lymphocytes, as well as thrombocytes. Both fractions were centrifuged, and the resulting pellets were resuspended in 350 µL RLT lysis buffer (Qiagen, Hilden, Germany) for RNA extraction and gene expression analysis.

### 4.2. RNA Isolation, cDNA Synthesis, and Quantitative PCR (qPCR) Analysis

RNA was isolated from the brain, gills, head kidney, trunk kidney, liver, muscle, and spleen of rainbow trout first with TRIzol (Thermo Fisher Scientific, Bremen, Germany) and then with the RNeasy Mini Kit (Qiagen), including an in-column DNase treatment for 30 min. For RNA isolation from cells, we used the ISOLATE II RNA Micro Kit (Bioline/Meridian Bioscience, Luckenwalde, Germany). RNA quantity was determined with a NanoDrop One^c^ (Thermo Fisher Scientific), and the integrity was assessed by agarose-gel electrophoresis. Total RNA was reverse-transcribed using a SensiFAST cDNA Synthesis Kit (Bioline/Meridian Bioscience).

The expression of (a) paralogues and ohnologues of *pias* genes in different tissues and cells of *Oncorhynchus* and (b) various immune genes in CHSE-214 cells transfected with *pias*-expressing vectors was profiled by establishing a panel of oligonucleotides (Table 2) using the Pyrosequencing Assay Design software (v.1.0.6; Biotage, Uppsala, Sweden) to amplify specific fragments between 95 and 195 bp.

The cDNA input into the individual RT-qPCR assays was equivalent to 2.5 ng total RNA isolated from cells and 75 ng total RNA isolated from tissues. The analyses were conducted with a LightCycler 96 instrument (Roche, Mannheim, Germany) using a SensiFAST SYBR No-ROX Kit (Bioline/Meridian Bioscience). Melting curve analyses validated the amplification of distinct products. In addition, we validated the size and quality of the PCR products on 1.5% agarose gels. Standard curves were generated based on the crossing points of 10-fold dilutions containing 10^3^ to 10^6^ copies of a PCR-generated standard fragment. The copy number was calculated for each fragment based on linear regression of the standard curve and relative to the amount of input RNA. Each expression value of the target genes was divided by the geometric mean of the reference genes *eef1a1* (eukaryotic translation elongation factor) [60] and *rps5* (ribosomal protein S5) [61].

### 4.3. Construction of Pias Expression Constructs

We amplified the open reading frames (ORF) of trout *pias1a1.1* (XM_036963708) and *pias2a1* (XM_036936540) using the oligonucleotide primers listed in Table 2. To this end, we performed standard PCRs using the Platinum Taq High-Fidelity DNA Polymerase (Thermo Fisher Scientific). The resulting amplicon was subcloned into pGEM-T Easy (Promega, Walldorf, Germany), retrieved by digestion with the restriction enzymes *Hind*III and *Eco*RI, and inserted into the mammalian expression vector v280 that had been previously double-digested with the above restriction enzymes. The resulting plasmids (v280_pias1, v280_pias2) were used for functional analyses.

We identified the subcellular localisation of Pias1 and Pias2 from rainbow trout by inserting the respective sequences in an expression vector flagged with green fluorescent proteins (GFP). In detail, we inserted a fragment coding for GFP at the 3‘-end of the CDS of the v280_pias1 and _pias2 plasmid. The GFP fragments had previously been amplified from commercial vectors (GFP: pAM505, NCBI-nucleotide accession code: AF140578) and inserted into the v280 clone [47,62] using the restriction sites for *Hind*III and *Eco*RI. The truncated *pias1* variant was produced by digesting the GFP-v280_pias1 plasmid with *Bam*HI to cut off 1357 bp of the downstream ORF. The two *Bam*HI restriction sites (GGATCA and GGATCC) were located at positions 609 to 611 in the ORF of *pias1* and immediately downstream of the GFP sequence. The ends of the linearised plasmid were subsequently joined using the T4 ligase (Promega).

### 4.4. Transfection, Luciferase Assay, and Confocal Microscopy

Endotoxin-free preparations (ZymoPure II Plasmid Maxi Prep Kit, ZymoResearch/Biozol, Eching, Germany) of the expression constructs for *pias1*, *pias2i*, and truncated *pias1* were transfected into CHSE-214 cells (Chinook salmon embryo-214; order ID: 91041114-1VL, Sigma-Aldrich/Merck) using the X-tremeGENE HP DNA Transfection Reagent (Roche). For co-transfection assays in six-well plates, we used 50 ng of the *ELAM* or the *mx* promoter constructs and increasing concentrations (50, 500, and 2000 ng) of the respective *pias*-expression construct. Three wells of each row were left as unstimulated controls, while the other three were additionally co-transfected with 50 ng of a vector coding for ifnγ [63]. The total DNA concentration of each transfection mixture was adjusted to 2500 ng/assay by adding the empty cloning vector. Finally, the luciferase activity of the cell lysates was measured with the Dual-Luciferase Reporter Assay System (Promega) with a Lumat LB9501 luminometer (Berthold, Bad Wildbad, Germany). Values were normalised against the protein concentration of the CHSE-214 cell extracts. Each transfection was assayed in triplicate; each transfection experiment was performed three times.

CHSE-214 cells transfected with the vector expressing GFP-tagged *pias1* or *pias2* from rainbow trout were fixed with 4% paraformaldehyde (Merck KGaA, Darmstadt, Germany) and subsequently inspected by confocal microscopy (LSM 780; Carl Zeiss Microscopy, Jena, Germany), equipped with a 63× oil-immersion DIC objective. For staining the nuclei, Hoechst 33342 dye (1 mg/mL; Sigma-Aldrich/Merck) was added to the medium 30 min before fixation.

### 4.5. Strategy for the Generation of a Pias-Knock-Out Cell Line

The genetically modified CHSE cell line CHSE-EC that stably expresses Cas9 and monomeric enhanced green fluorescence protein (mEGFP) [39] was chosen as the terminus a quo for this study. These cells were grown at 20 °C in Eagle Minimal Essential Medium with Earle’s salts (MEM) (Sigma-Aldrich) supplemented with 500 mg/mL G418 (Sigma-Aldrich), 30 mg/mL hygromycin (Thermo Fisher Scientific), 100 U/mL penicillin, 100 µg/mL streptomycin, 1% non-essential amino acids (NEAA; Biochrom AG), 2 mM L-glutamine (Biochrom AG), and 10% foetal bovine serum (FBS; Thermo Fisher Scientific).

Each sgRNA was designed on the first 100 nt of the coding sequence. The sgRNAs were synthesised using a 120 nt blunt-ended oligo (Sigma Aldrich/Merck) as a template and a RiboMAX Express T7 kit (Promega). The resulting product was purified using TRIzol (Thermo Fisher Scientific), then resuspended in RNAse-free and DNAse-free water and quantified with a NanoDrop One© (Thermo Fisher Scientific) before the transfection.

CHSE-EC cells were transfected with 100 ng mixed sgRNA (Table 2), together with 100 ng sgRNA targeting the mEGFP per 10 µL of cell suspension as previously described [39]. Transfected cells were plated onto a 25 cm^2^ flask and passaged weekly for 4 weeks. The mEGFP-negative cells were suspended in 2 mL MEM and sorted using a MoFlo XDP high-speed cell sorter (Beckman Coulter, Krefeld, Germany) with an incorporated air-cooled Coherent Sapphire laser (488 nm, 100 mW). The cells were sorted through a 70 μm nozzle at 60 psi in purify mode into 24-well plates and cultured with 1 mL MEM, weekly renewed for 4 months. The genomic DNA of CHSE-EC cells was isolated using DNeasy Blood & Tissue Kits (Qiagen) following the standard protocol. Sequencing primers (Table 2) were used to validate the success of the KO strategy.

### 4.6. Data Analysis

A parametric *t*-test or nonparametric Mann-Whitney U-test and GraphPad Prism software v.9 for macOSX were used to evaluate the statistical significance of the qRT-PCR data and reporter-gene measurements.

Alignment and phylogenetic reconstructions were performed to compare multiple Pias nucleotide and amino acid (aa) sequences using the ‘build’ function of ETE3 v3.1.1, as implemented on the GenomeNet site (https://www.genome.jp/tools/ete/, accessed on 1 March 2021) [64]. The tree was constructed using fasttree (with slow NNI and MLACC=3) to make the maximum-likelihood NNIs more exhaustive [65]. The gene synteny was determined using Genomicus v1.01 (https://www.genomicus.bio.ens.psl.eu/genomicus-100.01/cgi-bin/search.pl; accessed on 1 March 2021).

The three-dimensional structure was obtained using UCSF ChimeraX, offered as free software (http://www.rbvi.ucsf.edu/chimerax, accessed on 1 March 2021) [66]. Signal peptides were predicted using SignaIP-5.0 (http://www.cbs.dtu.dk/services/SignalP/ accessed on 1 March 2021). Disordered protein regions were predicted using PrDOS [67] (http://prdos.hgc.jp/cgi-bin/top.cgi accessed on 1 March 2021). An upstream analysis was performed using the Ingenuity program (Ingenuity Pathway Analyses/Qiagen accessed on 1 June 2021) to evaluate the target genes of STAT/NF-κB-dependent signalling.

## Figures and Tables

**Figure 1 ijms-22-12815-f001:**
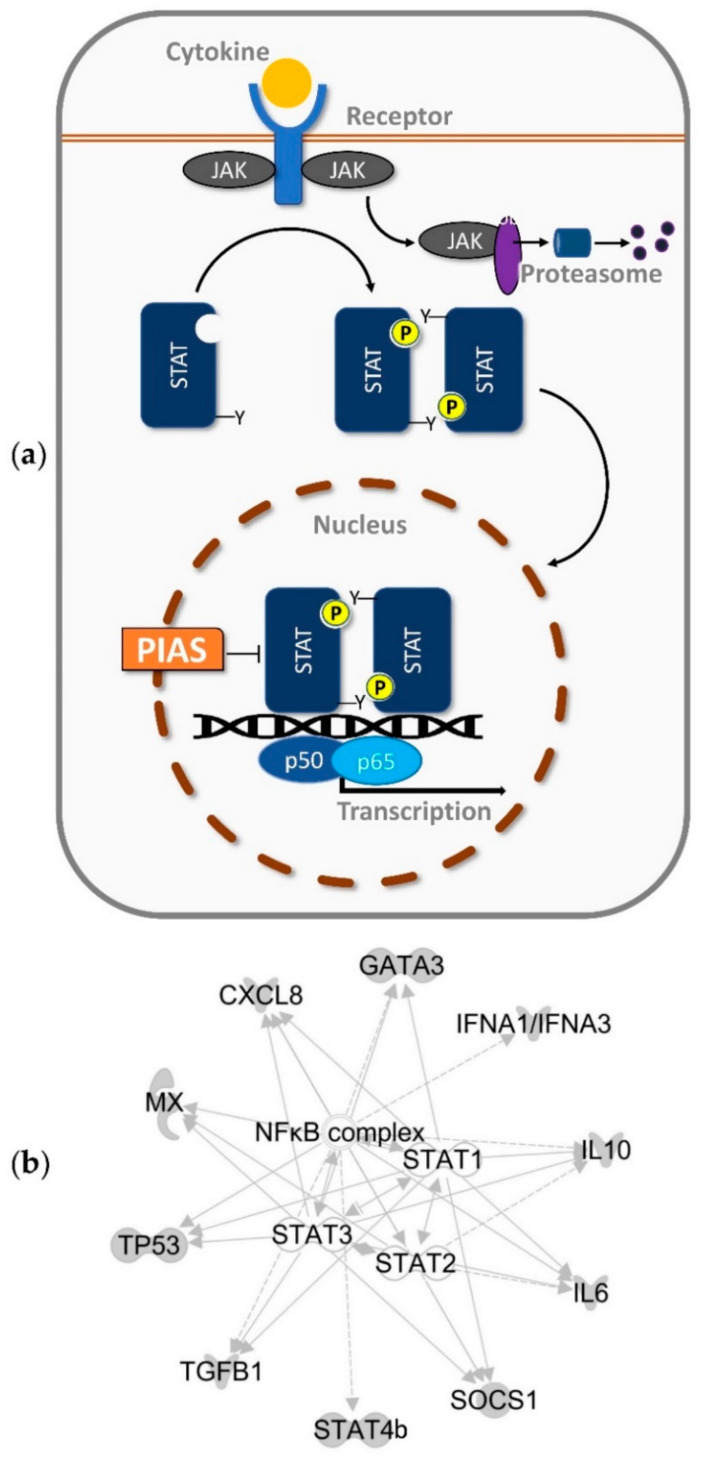
(**a**) Regulation of STAT/NF-κB-mediated pathways via PIAS; (**b**) target genes of the STAT/NF-κB-dependent signalling.

**Figure 2 ijms-22-12815-f002:**
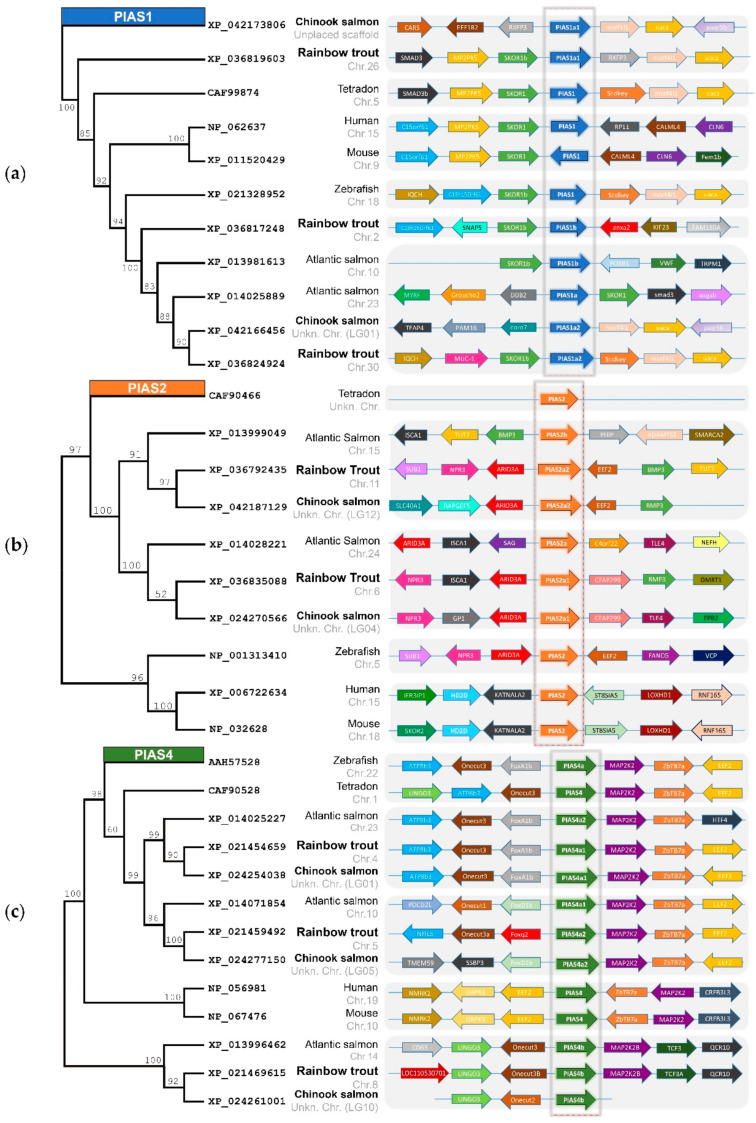
Phylogenetic relationship and synteny between the (**a**) *PIAS1*, (**b**) *PIAS2*, and (**c**) *PIAS4* genes from different vertebrate species. The bootstrap values of the phylogenetic analysis are given at the nodes of the tree. The NCBI protein accession codes, species names, and chromosomal location are listed between the phylogenetic and synteny analyses; the target species are labelled in bold. Arrows represent the reading direction of genes found in synteny; the same colours indicate orthologous genes. The figure is not scaled.

**Figure 3 ijms-22-12815-f003:**
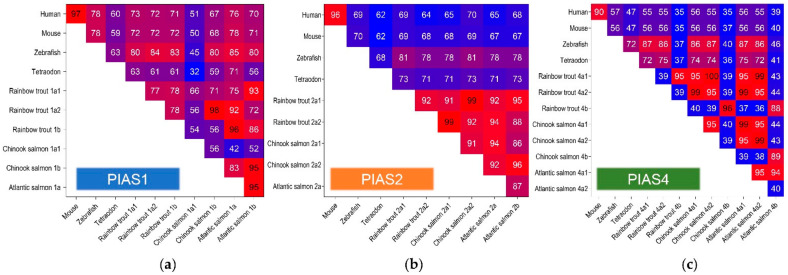
Sequence identity of (**a**) *PIAS1*, (**b**) *PIAS2*, and (**c**) *PIAS4* genes from different vertebrate species, listed to the left and below the individual graphs (different *PIAS* gene variants are indicated behind the species name). For NCBI protein accession codes, please refer to Figure 2.

**Figure 4 ijms-22-12815-f004:**
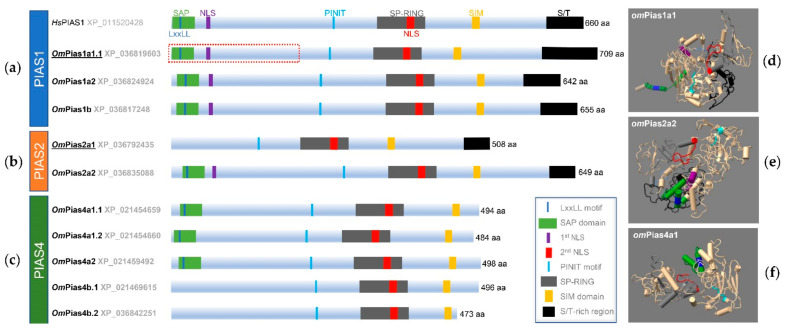
Representation of the domains and motifs characteristic of the variants of (**a**) Pias1, (**b**) Pias2, and (**c**) Pias4 in rainbow trout (Om). A schematic structure of the human (Hs) PIAS1 protein is included. The tertiary structures of (**d**) Pias1a1, (**e**) Pias2a2, and (**f**) Pias4a1 from rainbow trout were drawn using UCSF ChimeraX. The domains and motifs (**a**–**f**) are labelled according to the legend to the right of the Pias4 structures. The two underlined Pias variants were overexpressed in a cell model; the segment framed in red was overexpressed as a third ‘truncated Pias1′ variant.

**Figure 5 ijms-22-12815-f005:**
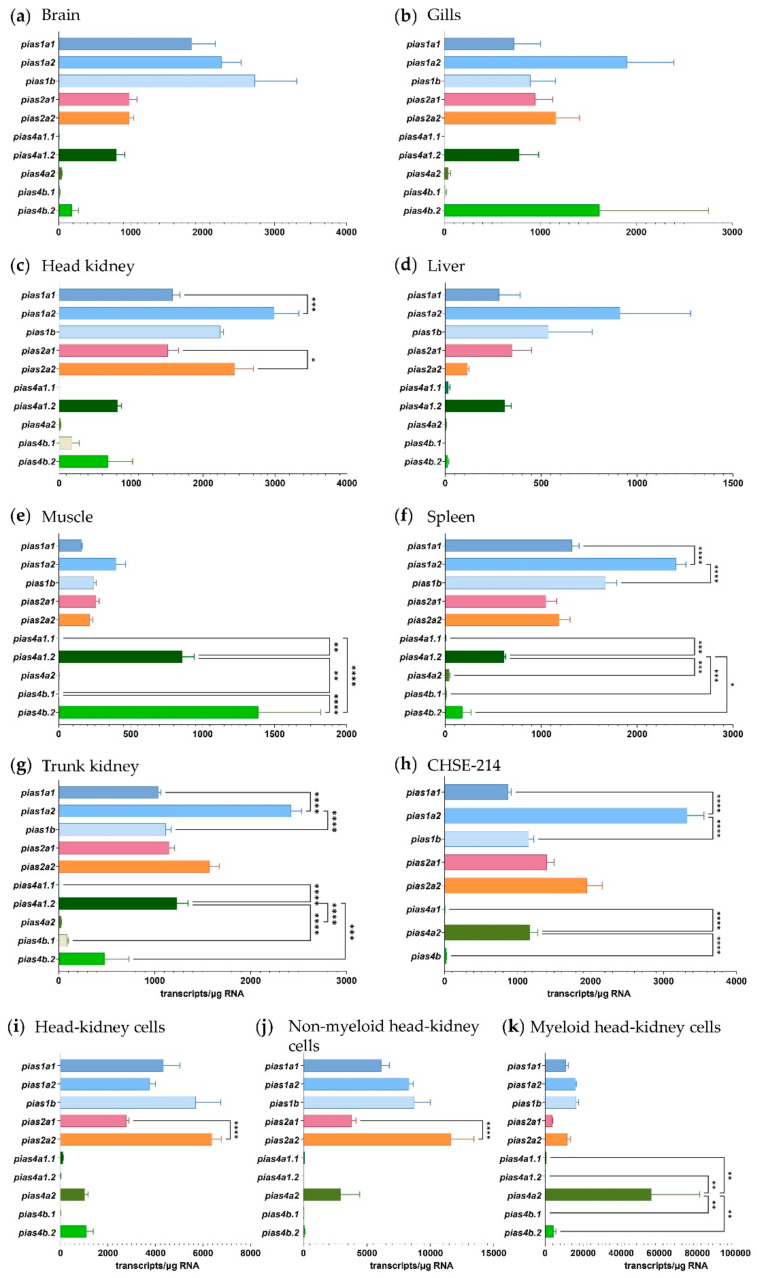
Levels of *pias* transcripts in rainbow trout tissues and salmonid cells (as listed above the diagrams). Bars represent the averaged copy numbers (*n* = 4) normalised against two reference genes; error bars represent the standard error of the mean. Asterisks represent significantly different transcript levels across ohnologues and transcript variants (*, *p* < 0.05; **, *p* < 0.01; ***, *p* < 0.001; ****, *p* < 0.0001).

**Figure 6 ijms-22-12815-f006:**
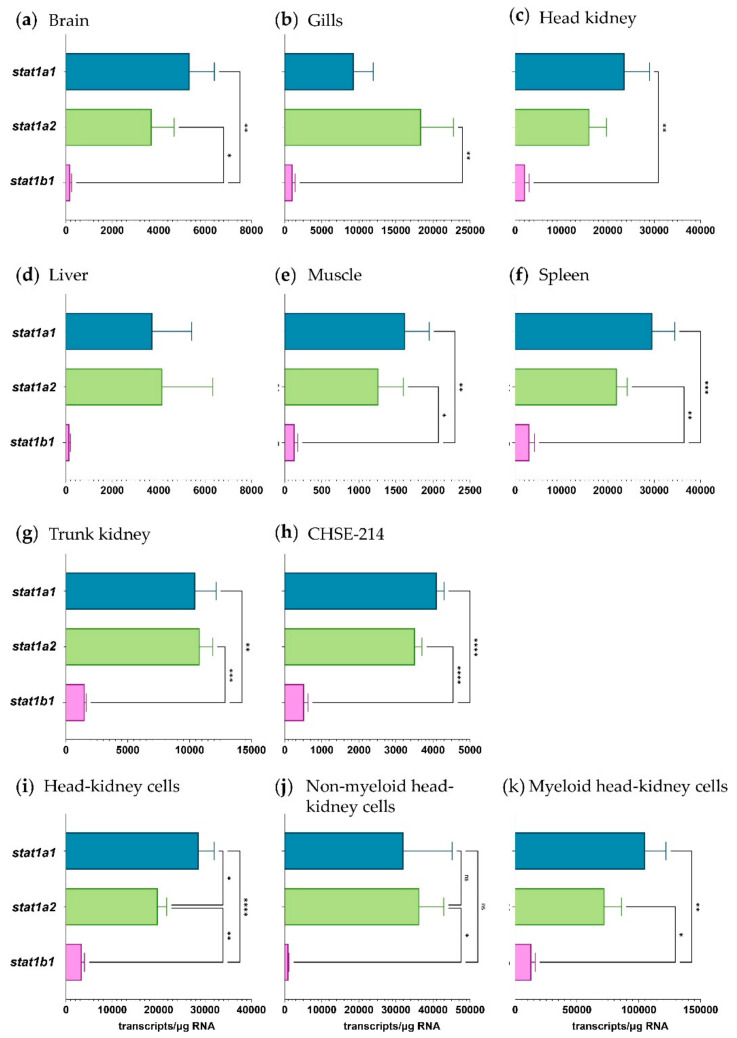
Expression profile of the *stat* genes in rainbow trout tissues and salmonid cells (as listed above the diagrams). Bars represent the averaged copy numbers (*n* = 4) normalised against two reference genes; error bars represent the standard error of the mean. Asterisks represent significantly different transcript levels (*, *p* < 0.05; **, *p* < 0.01; ***, *p* < 0.001; ****, *p* < 0.0001).

**Figure 7 ijms-22-12815-f007:**
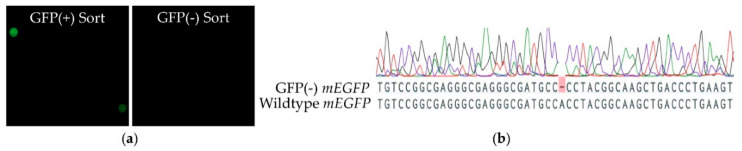
Knock-out targeting *pias1* and *pias2* in CHSE-EC cells. (**a**) Sorting of GFP-positive (+) and GFP-negative (-) cells; (**b**) genotype of GFP(-) clones.

**Figure 8 ijms-22-12815-f008:**
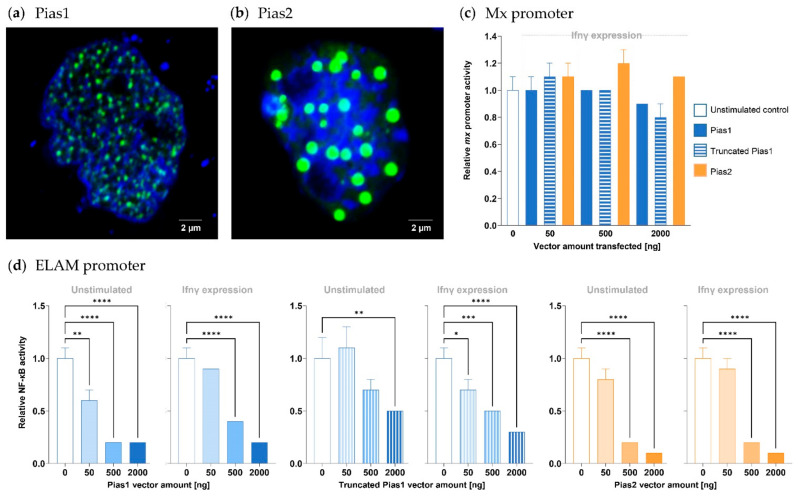
Overexpression of Pias factors in salmonid cell models. Confocal analysis of (**a**) GFP-tagged Pias1 (green) and (**b**) GFP-tagged Pias2 (green) in CHSE-214 cells; nuclei were stained with Hoechst 33342 dye (blue). The white scale bar represents 2 μm. The luciferase activity of CHSE-214 cells co-expressing (**c**) *mx*-reporter construct or (**d**) *ELAM*-reporter construct was determined in unstimulated control cells or ifnγ-expressing cells (as indicated above the graphs) co-expressing increasing concentrations of the *pias*-expressing vector (indicated on the abscissa). The luciferase activity in all cell cultures not expressing *pias* was set to 1.0. Statistical significance compared with the control group was assessed using one-way ANOVA (*, *p* < 0.05; **, *p* < 0.01; ***, *p* < 0.001; ****, *p* < 0.0001). The standard error of the mean (SEM) is indicated.

**Figure 9 ijms-22-12815-f009:**
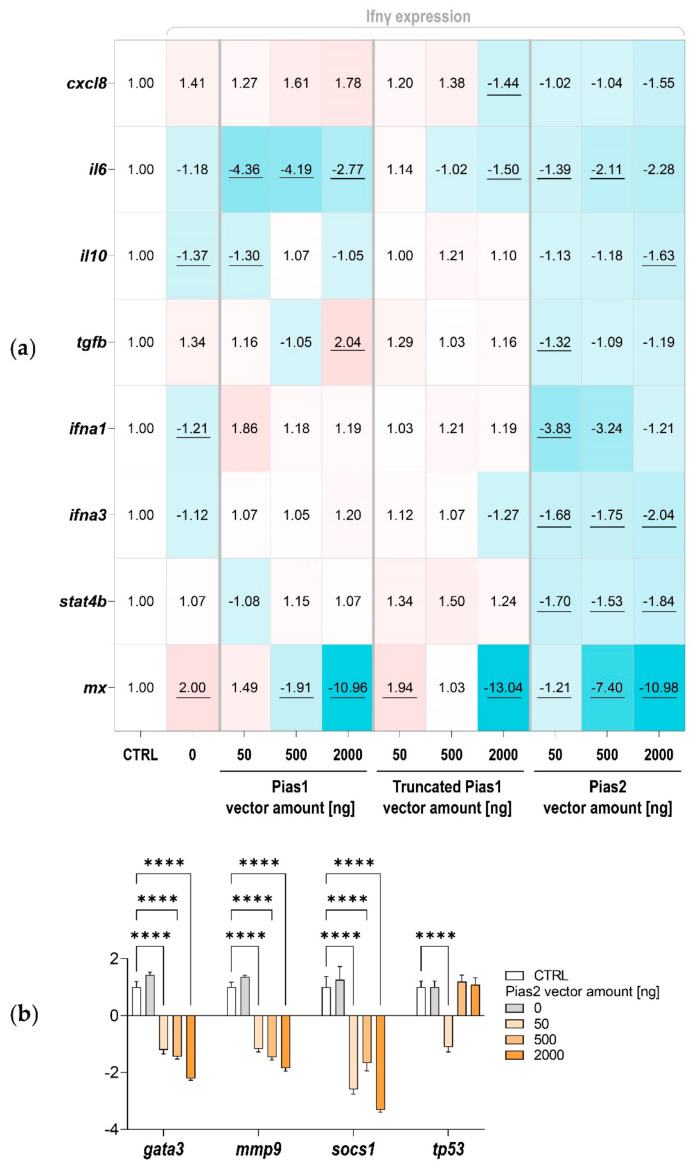
Expression profiling of *pias*-overexpressing CHSE-214 cells. (**a**) The heatmap and (**b**) bar chart illustrate the averaged fold-change values of the mRNA concentrations measured in cells transfected without (CTRL, set as 1.0) or with the *ifnγ*-expression vector (0 ng pias vector amount) together with *pias*-expression vectors (as indicated). The quantified transcripts are listed as gene symbols. All expression values shown in (**a**,**b**) were normalised against the geometric mean of two reference genes. Significantly different FC values compared to CTRL are underlined in (**a**). Statistical significance compared with the control group was assessed in (**b**) using one-way ANOVA (****, *p* < 0.0001); standard error of the mean (SEM) is indicated.

**Table 1 ijms-22-12815-t001:** *Pias* sequences identified in rainbow trout.

Nucleotide NCBI Accession Number	Chromosome	Gene	Transcript Isoform	CDS Length [nt]	UTR Length [bp] *5′* *3′*		Instability Motifs	Protein Length [aa]	Protein NCBI Accession Number
*Pias1*									
XM_036963708	26 (6) *	LOC110527003	pias1a1.1 (X1)	2130	41	1113	1 (3′ UTR)	709	XP_036819603
XM_036963709	26 (6)	LOC110527003	pias1a1.2 (X2)	2112	41	1113	1 (3′ UTR)	703	XP_036819604
XM_036961353	2	pias1b	pias1b	1968	78	3326	20 (3′ UTR)	655	XP_036817248
XM_036969029	30 (4)	LOC110521158	pias1a2	1929	140	1718	6 (3′ UTR)	642	XP_036824924
*Pias2*									
XM_036936540	11	pias2	pias2a2.1 (X1)	1965	289	3152	14 (3′ UTR)	654	XP_036792435
XM_036936542	11	pias2	pias2a2.2 (X2)	1950	286	3134	14 (3′ UTR)	649	XP_036792437
XM_036936543	11	pias2	pias2a2.3 (X3)	1920	411	3134	14 (3′ UTR)	639	XP_036792438
XM_036979193	6	LOC110525143	pias2a1.1 (X1)	1527	572	2502	2 (5′ UTR), 3 (3′ UTR)	508	XP_036835088
XM_036979194	6	LOC110525143	pias2a1.2 (X2)	1512	572	2502	2 (5′ UTR), 3 (3′ UTR)	503	XP_036835089
*Pias4*									
XM_021598984	4	LOC110521438	pias4a1.1 (X1)	1485	125	2933	24 (3′ UTR)	494	XP_021454659
XM_021598985	4	LOC110521438	pias4a1.2 (X2)	1455	60	2931	24 (3′ UTR)	484	XP_021454660
XM_021603817	5	LOC11052429	pias4a2	1497	132	3617	26 (3′ UTR)	498	XP_021459492
XM_021613940	8	pias4b	pias4b.1 (X1)	1491	410	696	1 (5′ UTR), 5 (3′ UTR)	496	XP_021469615
XM_036986356	8	pias4b	pias4b.2 (X2)	1422	498	696	1 (5′ UTR), 5 (3′ UTR)	473	XP_036842251

* Brackets indicate the former chromosomal location.

**Table 2 ijms-22-12815-t002:** Primers used in this study.

Gene Symbol	Primer Sequence 5′→3′ (Sense, Antisense)	Nucleotide NCBI Accession Number Used for Primer Design		Fragment Length [bp]
		*O. mykiss*	*O. tshawytscha*	
	Quantitative PCR Analysis (*Oncorhynchus mykiss*, *O. tshawytscha*)			
*Pias1a1.1*	GTTGGAAGGCACCTTCTGTGTT, CTACGGTCCAAAGGCATCAGG	XM_036963708	XM_031812568	108
*Pias1b*	GGAGCTACTCTATGGCGGTGT, ATCAGGAACCCAGACCATTCCA	XM_036961353	XM_042317872	99
*Pias1a2*	TAGGCAGGAATTTCTCCATGGC, AGAGAAGTTAACAGCTGACCCG	XM_036969029	XM_042310862	140
*Pias2a1.1*	GTGTGCATCTCCAGGGACTTTT, CTAAGAATGGAGTGGAACAGAAG	XM_036979193	XM_024414775	195
*Pias2a2.1*	GAGCTACGGAGCATGGTGTCA, AACTTTATCGACGCCGCTATCC	XM_036936540	XM_042331195	185
*Pias4a1.1*	ATTGGAAGCAGAGAACCGTCGA, ATTTTCGGGTGTCTGACCTGCA	XM_021598984	XM_024398270	158
*Pias4a1.2*	GCCTGCTAGGCTGGGAAACTA, CGCAGTAAAAGTGGTCTGAAGC	XM_021598985	―	99
*Pias4a2*	AGGAGGAGGGGGGAGGAGG, CGGACTGACCCCACAAACTGA	XM_021603817	XM_024421382	144
*Pias4b.1*	ACATAGCAGAAGCAATTAGGTTGT, AATCTGCTGGTGAGGGCAGTG	XM_021613940	XM_024421383	146
*Pias4b.2*	ACAAAGGCCCCGGAGTGAACA, GGGAGGGGAGTCAAGCTACAT	XM_021613941	―	129
*Stat1a1 (stat1-1)*	GAGAGCATCGACTGGGAAAATGT, AAACAACTTCCTGCTACAACACAA	NM_001124707	XM_024426102	131
*Stat1a2 (stat1-2)*	CCCCGTTCACATGGCCATGAT, CATAGAGACCGACAGAGAAAACA	XM_021608237	XM_042324083	95
*Stat1b1 (stat1ab)*	GGCCATGATAATCTGTAACTGTC, ACGTTAAAGACCTGAGGAACCG	XM_021579196	XM_042306329	150
*Cxcl8*	ATATAACACTTGTTACCAGCGAGA, ATTACTGAGGAGATGAGTCTGAG	HG917307	XM_024415648	106
*Il6*	GTGTTAGTTAAGGGGAATCCAGT, CCTTGCGGAACCAACAGTTTGT	NM_001124657	XM_024404411	128
*Il10*	TGCCCAGTGCAGACGTGTACC, TACACCACTTGAAGAGCCCCG	NM_001245099	XM_042324963	137
*Tgfb*	ATCAGGGATGAACAAGCTGAGG, CGGAGAGTTGCTGTGTGCGAA	XM_021591332	XM_024397891	161
*Ifna1*	TTGAAGAGAGCAAATGTATGATGG, TCCTGTACAGCCTACAGTTCATT	XM_024434105 (representative for *LOC110538045*, *LOC110538046*, *LOC110538047*,*LOC110538053*, *LOC110538937*, *LOC118937709)*	XM_024434105 (representative for *LOC112259401*, *LOC112259404*, *LOC121847201*, *LOC121847202*, *LOC112258510*)	173
*Ifna3*	CCAACATCACTTTACAGACACATA, GGGACAAGAAAAACCTGGACGA	XM_024432928 (representative for *LOC110511235*, *LOC110517168*, *LOC110538043*, *LOC110538058*)	XM_024389910 (representative for *LOC121838839*, *LOC112225816*, *LOC112258507*, *LOC112258508*)	140
*Stat4b*	ACCTCATCAAAAGCTCCTTTGTG, TTCACCACCAAAGTCAGATTGCT	XM_024388828	XM_024388828	112
*Mx*	GTAGCGGTATTGTAACACGATGC, TCGTGAAGCCCAGGATGAAATG	XM_036958922	XM_024415949	158
*Gata3*	CCACCTCCTCCACATAGTAGTC, GACCTGCCGGGGAACCGTG	XM_036957437	XM_042311479	160
*Mmp9*	TGCCAAGATAGAGGCTACAGTC, TGTCTTGGACCCATAGAGATAGT	XM_036986917	XM_024376362	181
*Socs1*	ACGGATTCTGCGTCGGAAAATAT, ACACAGTTCCCTGGCATCCGT	XM_036973400	XM_042313671	91
*Tp53*	GAATTTGAACCTGGTGGCAGTTC, CACCTCAAACAGACTCGGATCA	NM_001124692	XM_024394883	115
	Construction of PIAS-expression constructs			
*Pias1a1*	ATGCAAGCTTATGGCGGAGAGTGCGGAACT, CATACCAGACGTGATCTCGTTAGACGAATTCGCAT	XM_036963708		1968
*Pias2a1*	ATGCAAGCTTATGATCCTGACAAGAAAAATGGCGG, ACATCATCTCAGACATCATCTCATTGGACGAATTCGCAT	XM_036936540		1965
	SgRNA target sequence			
*Pias1a2*	AGTAACACTTGTAGCTCTGA GCGTAGCCTAGTAACACTTG TGTTTGTTGCGTCCTGCGTA		XM_024407180	20
*Pias1b*	TCTACAATAACACAAAAAGA ACACGACTCTGCAAGAGGGT TCTGTCAATCCATCTACAAT		XM_024410569	20
*Pias2a1.1*	ACACGTCGTAGAACGGGAGA GAGGGATGAGAGGGGCGGGC CCAGCAGCCCGCCCCTCTCA		XM_024414775	20
*Pias2a2*	CCATTTTTCTTGTCAGGATC GCGAAGCCCAGTAACACTTG TAGCTCCTCAAATTCCGCCA		XM_024407486	20
*mEGFP*	GGCGAGGGCGATGCCACCTA		[39]	20
	Sequencing primers for GFP(-) cells			
*Pias1a2*	TTAGTTGTTCTATTCGTGTGTCCTA, TTACACACACTGTGTGTACAAAACA		XM_024407180	500
*Pias1b*	GACCCCACTGCCTTTGTTTCAAACC, CATTCCTCCAAGGAGACAACCACCAG		XM_024410569	500
*Pias2a2*	AGTCTAAGCTTGACATCCATGAAAG, GTGTAGGCATTGGCTTAGCAATGC		XM_024407486	360
*Pias2a1.1*	CCCAAGGCGGTAGACAGTAGTCT, ACTGGGCTTTATGTTTCTGGTGACG		XM_024414775	500
*mEGFP*	ATGGTGAGCAAGGGCGAGCTG, GTCCTCCTTGAAGTCGATGCCCT		[39]	500

## Data Availability

The *pias*-cDNA sequences and associated metadata have been submitted to the ‘European Nucleotide Archive’ under accession number PRJEB47768.

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
