# Peer review of "PIAS Factors from Rainbow Trout Control NF-κB- and STAT-Dependent Gene Expression"

_ijms, 2021, doi:10.3390/ijms222312815_

Round 1
Reviewer 1 Report
In this study, Fabio Sarais and colleagues reported that PIAS factors from rainbow trout regulate the NF-κB- and STAT-dependent gene expression. They comparatively studied the structure, expression and function of the three PIAS orthologs in salmonid fish, and found all three PIAS factors can quench the basal activity of an NF-κB promoter in a dose-dependent fashion. The overexpression of all three Pias variants from trout showed strongly inhibited the transcript level of the antiviral Stat-dependent mx gene in ifnγ-expressing CHSE-214 cells. This study provides evidence for the multiplication of pias genes and their sub-functionalisation during salmonid evolution, and is of interest. But there are some major concerns need to be further checked.
Major:
- At line 164-215 in “2.2. The expression of pias genes is tissue- and cell-type-specific” part, why did the authors not detect the expression of pias gene in intestine and skin? They are also very important mucosal immune organs.
- In the Abstract section and at line 233, authors state that “knock-out of both pias1 and pias2 genes failed……” and “The knock-out of Pias factors in a CHSE cells using CRISPR/Cas9 technology failed, indicating the vital importance of the trout Pias factors.”, please discuss in detail the causes of knockout failure and why the knockout failure shows that pias gene is very important.
- At line 250-253 and in figure 8c, d, authors state that “The endogenous ifnγ synthesis induced a significant 4-fold increase in the NF-κB promoter activity…… the ifnγ-expression construct (set as 1.0 in Figure 8c, d).”, however, according to the result data, no significant difference showed in the NF-κB and mx promoter activities between the control group and the ifnγ-expressing group. Please
- At line 391-410 in “4.1. Sampling and cell sorting” part, authors state that “……the depleted (mAb21-negative) faction comprised of mostly B- and T-lymphocytes as well as thrombocytes.”, why did the authors not use Ficoll-Hypaque density-gradient centrifugation to separate leukocytes? After the myeloid cells are sorted out by magnetic beads, there are still non-lymphocytes and red blood cells, which cannot show that the rest are lymphocytes and thrombocytes. Therefore, the results of T/B lymphocyte gene expression analyzed by the authors may not be accurate. Please
- As stated by the authors, the results in this manuscript are obtained on CHSE cell line. It will be more convincing to investigate the role of PIAS factors in vivo and trout immune cell.
Minor:
- Line 28, “NF-κB-dependant” correct to “NF-κB-dependent”.
- At line 295, the “Two thousand nanogram……”, the format is inconsistent with the previous format, please revise.
Author Response
Reviewer 1: Thank you very much for your friendly evaluation of our manuscript.
At line 164-215 in “2.2. The expression of pias genes is tissue- and cell-type-specific” part, why did the authors not detect the expression of pias gene in intestine and skin? They are also very important mucosal immune organs.
Response: Yes, you are absolutely right. Especially the intestinal tract is a very interesting region from an immunological point of view. Unfortunately, we could not detect anything in this tissue, as everything had been used in a parallel study, which is in preparation at the moment. There we investigate the transcriptional response of mAb21-positive and -negative cells from gut to novel immune stimulants. However, in previous and future studies, we (will) include(d) skin and intestine (again) (see also doi:10.1016/j.fsi.2011.09.004 or doi:10.1016/j.fsi.2013.11.00).
In the Abstract section and at line 233, authors state that “knock-out of both pias1 and pias2 genes failed……” and “The knock-out of Pias factors in a CHSE cells using CRISPR/Cas9 technology failed, indicating the vital importance of the trout Pias factors.”, please discuss in detail the causes of knockout failure and why the knockout failure shows that pias gene is very important.
Response: We understand your concerns and therefore rephrased our conclusions from the failed knockout in a more cautious way. It is still plausible that PIAS is a vital gene family, as a search at the depmap portal (listing KO cell lines for cancer research) evaluated PIAS as “strongly selective”. We moved this indication to the Results section (lines 236-239) and deleted our speculations from the Abstract (line 21,22)+ Discussion (line 361,362). We hope this meets your justified criticism.
At line 250-253 and in figure 8c, d, authors state that “The endogenous ifnγ synthesis induced a significant 4-fold increase in the NF-κB promoter activity…… the ifnγ-expression construct (set as 1.0 in Figure 8c, d).”, however, according to the result data, no significant difference showed in the NF-κB and mx promoter activities between the control group and the ifnγ-expressing group.
Response: You were right; this was not clearly presented in our manuscript. The induction of the NF-κB and Mx promoter by ifnγ is a well reported observation (PMID: 11248675); however, it is a prerequisite to document the regulatory potential of the pias factors to reduce the promoter activity. To not confuse the reader, we only focused on this aspect and set the ifnγ-induced promoter activation as baseline. This is indicated now in line 261/262 and 269.
At line 391-410 in “4.1. why did the authors not use Ficoll-Hypaque density-gradient centrifugation to separate leukocytes? After the myeloid cells are sorted out by magnetic beads, there are still non-lymphocytes and red blood cells, which cannot show that the rest are lymphocytes and thrombocytes. Therefore, the results of T/B lymphocyte gene expression analyzed by the authors may not be accurate.
Response: We thank you for this comment. We used a Percoll gradient to prepare leukocyte suspensions before the magnetic sorting, in order to avoid the presence of erythrocytes. A better description of the protocol followed was included in the text, lines 412-416.
As stated by the authors, the results in this manuscript are obtained on CHSE cell line. It will be more convincing to investigate the role of PIAS factors in vivo and trout immune cell.
Response: Thank you for this advice; we added this aspect at the end of our conclusions section (line 403/404).
Line 28, “NF-κB-dependant” correct to “NF-κB-dependent”.
At line 295, the “Two thousand nanogram……”, the format is inconsistent with the previous format, please revise.
Response: Thank you very much for careful reading. We corrected the respective issues and consulted, in addition, a professional proofreading service.
Reviewer 2 Report
The study by Sarais and colleagues evaluate pias genes in salmonids. The authors described multiple copies of pias genes identified in rainbow trout, and examined their phylogenetic relationship, key features and architecture. They also analyzed expression of these genes in various tissues and cell types. Finally, they performed over-expression experiments and attempted to knock-out Pias factors using CRISPR/Cas9 technology. Together, this offers meaningful insights into these important regulatory factors in salmonids. Despite these strengths, there are a few questions/comments that prevent me from recommending publication of this manuscript in its current form.
- Figures 5i, 5k, and 5l show the abundance of pias gene transcripts in total cells, lymphoid and myeloid cells from the head kidney, respectively. The monoclonal antibody mAB21 was used in the discrimination of cellular subsets. Those that were mAB21 positive were defined as myeloid cells, consistent with previous work by members of this group. However, it is unclear why mAB21 negative cells were defined as lymphoid cells. Were other antibodies or stains used to define these cells? A significant fraction of cells in the head kidney are early hematopoietic progenitors at various early stages of differentiation. Is there a possibility that a proportion of those cells currently labeled as “lymphoid” actually represent these early hematopoietic cell types? If so, it may be more informative to capture this in the descriptions since the contributions(s) of these pias genes may be to hematopoiesis and not lymphoid function. Proliferation, survival and lineage-specific differentiation would offer viable possibilities.
- A minor note related to figures 5i, 5k, and 5l. There appears to be a panel missing. Alternatively, the current panels may be mislabeled.
- "Altogether, this suggests that the CRISPR/Cas9 system basically worked in CHSE-EC cells, but the deletion of the genes of interest may have been lethal for the cell [8]." This conclusion has potentially significant implications. Was there any complementary support from other sources pointing to lethality among cells that are unable to express pias genes? Only a brief mention of in silico analyses currently appears in the Discussion section.
- Although the article is largely devoid of grammatical and spelling mistakes, there are still a number of cases throughout the manuscript. Please revise.
Author Response
Reviewer 2: Thank you very much for the warm perception of our study.
- Figures 5i, 5k, and 5l - it is unclear why mAB21 negative cells were defined as lymphoid cells. Were other antibodies or stains used to define these cells? A significant fraction of cells in the head kidney are early hematopoietic progenitors at various early stages of differentiation. Is there a possibility that a proportion of those cells currently labeled as “lymphoid” actually represent these early hematopoietic cell types? If so, it may be more informative to capture this in the descriptions since the contributions(s) of these pias genes may be to hematopoiesis and not lymphoid function. Proliferation, survival and lineage-specific differentiation would offer viable possibilities.
Response: We thank you for this valuable comment and agree with your suggestion. "Lymphoid fraction" was exchanged in the text for "non-myeloid fraction" (see lines 183-185 and ff.) Although we currently have antibodies anti- trout thrombocytes, anti-IgM, anti-CD8 and an anti-trout pan T cells, to partially analyze tissue cell composition and immunophenotype, we cannot rule out the presence of hematopoietic progenitors in this cell mixture. Currently, there are no anti- trout CD34 or anti-trout ckit antibodies available. Your suggestion regarding pias function in hematopoiesis was incorporated and discussed in the text (lines 349-353).
- A minor note related to figures 5i, 5k, and 5l. There appears to be a panel missing. Alternatively, the current panels may be mislabeled.
Response: Thank you for your careful review - we corrected the labeling of the figures 5 and 6.
- "Altogether, this suggests that the CRISPR/Cas9 system basically worked in CHSE-EC cells, but the deletion of the genes of interest may have been lethal for the cell [8]." This conclusion has potentially significant implications. Was there any complementary support from other sources pointing to lethality among cells that are unable to express pias genes? Only a brief mention of in silico analyses currently appears in the Discussion section.
Response: The second reviewer’s report mentions similar concerns as you (see above). Therefore, we moved the in-silico indication to the Results section and deleted our speculations from the Discussion and the Abstract (cf. lines 236-239; 21,22; 361,362).
- Although the article is largely devoid of grammatical and spelling mistakes, there are still a number of cases throughout the manuscript. Please revise.
Response: The article has been proofread by a professional service.